# Novel AIEgen-Functionalized Diselenide-Crosslinked Polymer Gels as Fluorescent Probes and Drug Release Carriers

**DOI:** 10.3390/polym12030551

**Published:** 2020-03-03

**Authors:** Jie Zhao, Xiangqiang Pan, Jian Zhu, Xiulin Zhu

**Affiliations:** 1State and Local Joint Engineering Laboratory for Novel Functional Polymeric Materials, Jiangsu Key Laboratory of Advanced Functional Polymer Design and Application, Department of Polymer Science and Engineering, College of Chemistry, Chemical Engineering and Materials Science, Soochow University, Suzhou 215123, China; 20174209252@stu.suda.edu.cn (J.Z.); xlzhu@suda.edu.cn (X.Z.); 2Global Institute of Software Technology, Suzhou 215163, China

**Keywords:** polymer gels, redox response, aggregation-induced emission (AIE), fluorescent probes, drug release carriers

## Abstract

Stimuli-responsive functional gels have shown significant potential for application in biosensing and drug release systems. In this study, aggregation-induced emission luminogen (AIEgen)-functionalized, diselenide-crosslinked polymer gels were synthesized via free radical copolymerization. A series of polymer gels with different crosslink densities or tetraphenylethylene (TPE) contents were synthesized. The diselenide crosslinker in the gels could be fragmented in the presence of H_2_O_2_ or dithiothreitol (DTT) due to its redox-responsive property. Thus, the TPE-containing polymer chains were released into the aqueous solution. As a result, the aqueous solution exhibited enhanced fluorescence emission due to the strong hydrophobicity of TPE. The degradation of polymer gels and fluorescence enhancement in an aqueous solution under different H_2_O_2_ or DTT concentrations were studied. Furthermore, the polymer gels could be used as drug carriers, suggesting a visual drug release process under the action of external redox agents. The AIEgen-functionalized, diselenide-crosslinked polymer gels hold great potential in the biomedical area for biosensing and controlled drug delivery.

## 1. Introduction

Stimuli-responsive materials can undergo relatively large and abrupt physical or chemical changes in response to small external stimuli [1,2]. In the last few decades, stimuli-responsive functional gels, responsive to light, temperature, pH, ionic strength, force, and redox reactions, among others, have attracted significant attention in sensing, drug delivery, and biotechnology [3,4,5,6,7,8,9,10,11]. Among these functionalities, redox-responsive polymer gels play an important role for application in physiological environments, where the redox process is constantly and widely present [12,13,14].

Fluorescent probes are highly efficient and sensitive bio-optical detectors that have demonstrated a significant value in bioimaging and biosensing applications [15,16,17]. Recently, fluorescent probes based on the aggregation-induced emission (AIE) effect have attracted great attention [18,19,20,21], especially tetraphenylethylene (TPE) has been extensively studied for its high quantum yield and facile synthesis. In 2016, Ishiwari et al. introduced TPE molecules into polyacrylic acid hydrogels, showing enhanced fluorescence emission after adding Ca^2+^ due to significant chain folding thereof [22]. Later, in the same year, Lei et al. designed temperature-responsive polymer gels, whose fluorescence switch was driven by a combination of poly(N-isopropylacrylamide) (PNIPAm) and the AIE effect [23].

Selenium-containing polymers showed versatile responsive behaviors to multiple stimuli, such as oxidation, reduction, and irradiation [24,25,26,27,28,29], which make them potentially useful as bio-building blocks. Redox responsiveness is an important property of diselenide-containing polymers [30,31,32,33]. Compared with the disulfide bond, the lower binding energy of the selenium bond (172 kJ mol^−1^) gives them a high sensitivity to oxidative and reductive stimuli. In 2010, Ma et al. reported the first redox-responsive block copolymer containing a diselenide functional group. The copolymer self-assembled into spherical micelles in water, which showed responsiveness to redox stimuli in a tumor microenvironment [34]. After that, more diselenide-containing drug delivery systems were developed, including micelles, hydrogels, and metal-organic frameworks (MOFs), in response to redox stimuli [35,36,37]. In 2018, Sun et al. prepared a multi-stimulated, responsive, biodegradable, diselenide-crosslinked, starch-based hydrogel for controlled drug delivery [38].

In this work, we designed and prepared novel aggregation-induced emission luminogen (AIEgen)-functionalized, diselenide-crosslinked polymer gels. The obtained gels could be degraded with redox stimuli due to the responsive behavior of the diselenide crosslinker. As a result, the TPE-containing polymer chains were released into an aqueous solution (Scheme 1), which exhibited enhanced fluorescence emission due to the strong hydrophobicity of TPE. Furthermore, the polymer gels were able to encapsulate drugs, such as doxorubicin (DOX), and function as drug carriers, suggesting a visual drug release process under the action of external redox agents. The AIEgen-functionalized and diselenide-crosslinked polymer gels showed great potential applications as biomedical materials.

## 2. Materials and Methods

### 2.1. Materials

Prior to use, acrylic acid (98%; Energy Chemical, Shanghai, China), 2-hydroxyethyl methacrylate (HEMA, 96%; Energy Chemical, Shanghai, China), and acrylchloride (AR; Macklin, Shanghai, China) were purified by passage through an Al_2_O_3_ column to remove inhibitors. γ-Selenobutyrolactone was synthesized according to a previously reported method [39]. 4-(1,2,2-Triphenylvinyl)phenyl acrylate (TPE-a) was synthesized according to a previously reported method [22]. 2,2’-Azoisobutyronitrile (AIBN, 98%; Sigma-Aldrich, St. Louis, MO, USA) was recrystallized from ethanol and then stored in a refrigerator at 4 °C. Benzophenone (CP; Sinopharm Chemical, Shanghai, China), 4-hydroxylbenzophenone (98%; Energy Chemical, Shanghai, China), zinc powder (99.99% trace metals basis, 600 mesh; Aladdin, Shanghai, China), titanium tetrachloride (TiCl_4_, AR; Enox, Changshu, China), 1,5,7-triazabicyclo(4.4.0)dec-5-ene (TBD, 97%; Energy Chemical, Shanghai, China), hydrochloric acid (HCl, AR; Enox, Changshu, China), triethylamine (TEA, AR; Shanghai Chemical Reagents, Shanghai, China) were used as received. Tetrahydrofuran (THF, AR; Enox), dimethylformamide (DMF, AR), methanol (MeOH, AR), acetone (AR), ethyl acetate (EA, AR), trichloromethane (CHCl_3_, AR) were purchased from Enox (Shanghai, China) and used without further treatment. Doxorubicin (DOX, 97%) was purchased from Aladdin (Shanghai, China).

### 2.2. Characterization

^1^H NMR and ^13^C NMR spectra were recorded on a Bruker Avance 300 spectrometer (Bruker Biospin International AG, Postfach, Switzerland). Chemical shifts were presented in parts per million (δ) relative to CHCl_3_ (7.26 ppm in ^1^H NMR). Fourier transform infrared spectroscopy (FT-IR) data were recorded with a Bruker TENSOR 27 FT-IR instrument (Bruker Optics, Billerica, MA, USA) using the conventional KBr pellet method. The elemental composition was measured with X-ray photoelectron spectroscopy (XPS) (ESCALAB 250 XI, Al KR source, Thermo Fisher Scientific, Waltham, MA, USA). The morphology of samples was observed via Hitachi SU8010 scanning electron microscopy (Hitachi High-Tech, Okinawa, Japan) with an operated voltage at 5kV. The fluorescence emission spectra (FL) were obtained on a HITACHI F-4600 fluorescence spectrophotometer (Hitachi High-Tech, Okinawa, Japan) at room temperature. Ultraviolet visible (UV-vis) absorption spectra were measured with a Shimadzu UV-2600 spectrophotometer (Shimadzu, Suzhou, China).

### 2.3. Synthesis of the Crosslinker (HEMA-Se)_2_

γ-Selenobutyrolactone (3.3 g, 20 mmol), 2-hydroxyethyl methacrylate (HEMA; 2.6 g, 20 mmol), and 1,5,7-triazabicyclo(4.4.0)dec-5-ene (TBD; 0.084 g, 0.6 mmol) were dissolved in THF and stirred at 50 °C overnight. After that, THF was removed by distillation under reduced pressure. The crude product was purified by column chromatography on a silica gel (ethyl acetate/petroleum ether (v/v = 1/10) to obtain (HEMA-Se)_2_ (0.94 g, 16%). The structure of the obtained compound was characterized by NMR. ^1^H-NMR (300 MHz, CDCl_3_): δ 6.12, 5.59 (s, 4H, **CH_2_**C(CH_3_)-), 4.34 (s, 8H, -O**CH_2_CH_2_**O-), 2.94–2.89 (t, 4H, -CH_2_CH_2_**CH_2_**-Se-), 2.49–2.44 (t, 4H, -**CH_2_**CH_2_CH_2_-Se-), 2.09–2.04 (m, 4H, -CH_2_**CH_2_**CH_2_-Se-), 1.94 (s, 6H, CH_2_ C(**CH_3_**)-) ppm. ^13^C-NMR (75 MHz, CDCl_3_): δ 172.61, 167.09, 135.92, 126.10, 62.38, 62.15, 33.47, 28.57, 25.90, 18.30 ppm. ^77^Se-NMR (114 MHz, CDCl_3_): δ 302.02 ppm.

### 2.4. Preparation of AIEgen-Functionalized and Diselenide Cross-Linked Polymer Gels SeSe_y_-PAA-TPE_x_

A DMF (2 mL) solution of a mixture of TPE-a (64 mg, 0.16 mmol), acrylic acid (0.55 g, 7.6 mmol), (HEMA-Se)_2_ (0.13 g, 0.24 mmol), and AIBN (13 mg, 0.08 mmol) was degassed by the standard freeze–pump–thaw method (at least 3 cycles). The mixture was allowed to stand at 70 °C for 3 h and then cooled to 25 °C. The resultant gelatinous material was subjected to Soxhlet extraction with a mixture of methanol/acetone (1/1, v/v) for 24 h, dried at 40 °C under reduced pressure for 48 h to obtain SeSe_0.03_-poly(acrylic acid)(PAA)-TPE_0.02_ as a light-yellow solid. A similar procedure was used to obtain SeSe_0.01_-PAA-TPE_0.02_ and SeSe_0.03_-PAA-TPE_0.05_ from TPE-a, acrylic acid, (HEMA-Se)_2_, and AIBN with the corresponding monomer feed ratios.

### 2.5. Oxidation Responsive Behaviors and Fluorescence Variation of Polymer Gels SeSe_y_-PAA-TPE_x_

The oxidation responsive behaviors of the SeSe_y_-PAA-TPE_x_ polymer gels were studied by immersing the dry gels in 0.01 wt.% H_2_O_2_, 0.001 wt.% H_2_O_2_, and 0.0001 wt.% H_2_O_2_ in phosphate buffer (PB) solution (1 M, pH = 7.4) at room temperature. SeSe_0.03_-PAA-TPE_0.02_ was used as an example and the mixture was dried for FT-IR and XPS tests.

### 2.6. Reduction Responsive Behaviors and Fluorescence Variation of Polymer Gels SeSe_y_-PAA-TPE_x_

The reduction responsive behaviors of the SeSe_y_-PAA-TPE_x_ polymer gels were studied by immersing the dry gels in 10 mM DTT, 1 mM DTT, and 0.1 mM DTT in PB solution (1 M, pH = 7.4) at room temperature. SeSe_0.03_-PAA-TPE_0.02_ was used as an example and the mixture was dried for FT-IR and XPS tests.

### 2.7. Drug Loading and Release Behaviors of Polymer Gels SeSe_y_-PAA-TPE_x_ under Redox Conditions

Doxorubicin (DOX) was chosen as the model drug to test the drug loading and release behaviors of the polymer gel SeSe_0.01_-PAA-TPE_0.02_. The following process was carried out in the dark. The dry gels (20 mg) were swollen in a DOX/PB solution (0.1 mg mL^−1^, 10 mL) at room temperature for 24 h. The DOX-loaded gels were taken out and rinsed with a small amount of PB solution (1 M, pH = 7.4). The collected supernatant was brought to a volume and the absorbance at 480 nm was tested. Afterwards, dry DOX-loaded gels were obtained by freeze drying for 12 h.

A calibration curve was obtained by measuring the absorbance at 480 nm of the DOX/PB solution (1 M, pH = 7.4) with different concentrations, so that the amount of DOX loaded onto the polymer gels could be determined. The drug-loading capacities (DLC) and drug-loading efficiencies (DLE) were calculated using the following equations:DLC (wt.%) = (weight of loaded drug/weight of (dry gel + loaded drug)) × 100%(1)
DLE (wt.%) = (weight of loaded drug/weight of drug in feed) × 100%(2)

Drug release studies in vitro were investigated in a PB solution (1 M, pH = 7.4) with/without 0.001 wt.% H_2_O_2_/10 mM DTT. The reaction mixture was stirred at 37 °C. Then, fluorescence spectrophotometry was used to monitor the change in fluorescence intensity of the reaction solution and UV-vis absorption spectra were used to monitor the release behaviors of the DOX-loaded gels.

### 2.8. Cytotoxicity Tests

The cytotoxicity of SeSe_0.01_-PAA-TPE_0.02_ after the reactions with 0.01 wt.% H_2_O_2_ or 10 mM DTT was evaluated using HeLa cells and the cholecystokinin (CCK) test. A control group (culture medium and cells) was set up to compare with the experimental group. The cells were incubated with samples in a range of concentrations from 0.05 to 0.8 mg mL^−1^ for 24 h. The viability of HeLa cells was then measured by the CCK-8 assay. The absorbance at 450 nm was recorded in a microplate reader (Thermo Fisher Scientific Inc.).

## 3. Results and Discussion

### 3.1. Synthesis of AIEgen-Functionalized Diselenide-Crosslinked Polymer Gels SeSe_y_-PAA-TPE_x_

The AIEgen-functionalized and diselenide-crosslinked polymer gels (SeSe_y_-PAA-TPE_x_) were synthesized via free radical copolymerization of aggregation-induced emission (AIE)-functionalized monomer 4-(1,2,2-triphenylvinyl)phenyl acrylate (TPE-a) with acrylic acid (AA) using (HEMA-Se)_2_ as the crosslinker. The synthetic route is shown in Scheme 1. The structures of two compounds were confirmed by NMR (Appendix A). Three polymer gels were prepared according to the method with different crosslink densities or TPE contents, which were labeled as SeSe_0.03_-PAA-TPE_0.02_, SeSe_0.01_-PAA-TPE_0.02_, and SeSe_0.03_-PAA-TPE_0.05_. Detailed information is shown in Table 1. The gels showed good swelling properties in PB solution with relationships to their structures, for example, the content of the crosslinker and the other two components. The dry gels, as light-yellow solids, showed a strong blue emission under UV lamp (365 nm) due to the introduction of TPE-a (Figure 1a). The structure of polymer gels was characterized using FT-IR and scanning electron microscope (SEM). As depicted in Figure 2b, the absorption peak at 800 cm^−1^ according to the vibrations of C–Se groups indicated the successful introduction of selenium moieties. The surface morphology of the polymer gels showed an irregular three-dimensional network structure (Figure 1b), which conformed to the characteristics of the random copolymer gels. The above FT-IR and SEM results confirmed that the AIEgen-functionalized and diselenide-crosslinked polymer gels (SeSe_y_-PAA-TPE_x_) were successfully prepared.

### 3.2. Oxidation and Reduction Responsiveness of Polymer Gels SeSe_y_-PAA-TPE_x_

Due to the unique redox-responsive cleavage of the diselenide linkage [30,34], the SeSe_y_-PAA-TPE_x_ polymer gels exhibited a redox-responsive character. When the dry gels were immersed into a H_2_O_2_ or DTT solution, the polymer chains with TPE were released into the solution due to degradation of the diselenide linkage in SeSe_y_-PAA-TPE_x_. The reaction solution showed an enhancement of fluorescence emission due to the strong hydrophobicity of TPE. The X-ray photoelectron spectroscopy (XPS) data (Figure 2a) displayed that, after the reactions with 0.01 wt.% H_2_O_2_, the binding energy of Se 3d^5^ shifted from 55.90 to 59.31 eV, which was very close to the valence of the seleninic acid group. Besides, after the gels were treated with 10 mM DTT, the binding energy of Se 3d^5^ decreased from 55.90 to 55.16 eV, which corresponded to the selenol group. These results indicated that the diselenide bond in the gels converted to seleninic acid after treatment with 0.01 wt.% H_2_O_2_ or converted to selenol after treatment with the DTT solution, which resulted in the degradation of the polymer gels. A comparison of the FT-IR spectra of SeSe_0.03_-PAA-TPE_0.02_ before and after the redox reaction (Figure 2b) showed that the absorption peak of the Se–C bond at 800 cm^−1^ disappeared after treatment, which indicated the transformation of the Se–C group. This transformation also resulted in the disappearance of the C=O vibration peak at 1728 cm^−1^ and slightly changed the vibration peaks of the CH_2_ bond in the Se–C group from 1452 cm^−1^ to around 1415 cm^−1^. These results confirmed the transformation of diselenide to seleninic acid or selenol groups after treatment with oxidative or reductive reagents.

The oxidation and reduction responsive behaviors of the three polymer gels under different H_2_O_2_ or DTT concentrations were studied by fluorescence emission spectra (Appendix A). When the dry gels were added to a H_2_O_2_ or DTT solution, the fluorescence intensity of the reaction solution increased significantly as H_2_O_2_ concentration increased from 0.0001 wt.% to 0.01 wt.% (Figure 2c) or DTT concentration increased from 0.1 mM to 10 mM (Figure 2e). These results confirmed that the diselenide crosslinker could be decomposed under both the oxidation and reductive conditions. An increase in the concentration of the oxidant or reductant resulted in the increasing degradation of the diselenide bond, which increased the density of fluorescence emission. As a result, SeSe_y_-PAA-TPE_x_ gels can be used as fluorescent probes for detecting redox agents. Through a comparative analysis, it was found that SeSe_0.01_-PAA-TPE_0.02_ showed a significant fluorescence enhancement in solution after being treated with 0.0001 wt.% H_2_O_2_ or 0.1 mM DTT. The results indicated that the lower the crosslink density, the lower the detection limit for redox agents. In addition, SeSe_0.01_-PAA-TPE_0.05_ showed higher fluorescence intensity in solution in the presence of 0.01 wt.% H_2_O_2_ or 10 mM DTT. Thus, the higher the TPE content, the stronger the fluorescence intensity when the same concentration of redox agents was detected, which provided a useful way to adjust the fluorescence intensity. In addition, the response time of the three polymer gels with different crosslink density or TPE content was basically the same, whereas there was a difference in their fluorescence intensities and response rates (Figure 2d,f).

The recovery of the polymer gels after oxidation or reduction was also studied. As shown in Figure 3a, fluorescence enhancement was observed only under oxidizing conditions when the selenium bonds could be oxidized to selenic acid [34]. For this solution, the polymer chain could be further folded after the addition of Ca^2+^ [40,41]. The diselenide bond also could be reduced to selenol under reductive conditions [34]. When Ca^2+^ was added, selenol groups were close to each other and reoxidized by air to selenium bonds (Figure 3b). Insoluble matter was formed, which decomposed in a strong acid. FT-IR spectra (Figure 3c) showed that the absorption peak located at 800 cm^−1^ appeared belonging to the vibrations of C–Se groups, and the other absorption peaks were also almost consistent before and after the recovery. Moreover, the fluorescence emission spectra (Figure 3d) proved the successfully recovery of the reduced product. 

### 3.3. Drug Release Behavior of Polymer Gel SeSe_y_-PAA-TPE_x_ under Redox Conditions

The redox process is constantly and widely present in physiological environments; normal cells keep a lower level of redox agents, while the redox process in tumor cells would increase significantly [12,13,14]. The polymer gels can be used in responsive to redox stimuli in the tumor microenvironment as a targeted delivery system. Due to the oxidation and reduction responsive cleavage of the diselenide bond, the use of the SeSe_y_-PAA-TPE_x_ polymer gels for loading and releasing functional molecules such as drug carriers, looks promising. For this purpose, controlled release experiments under oxidation and reduction stimuli were conducted and SeSe_0.01_-PAA-TPE_0.02_ was chosen as the drug carrier. According to redox response data, we could easily adjust the drug-loading capacities and release behaviors by changing the crosslink density or TPE content. Here, we focused on the fluorescence-enhancing behavior in solution for the fluorescent polymer gel probes during redox-triggered drug delivery. The anticancer drug doxorubicin (DOX) was chosen as the model molecule for encapsulation, which showed a strong UV-vis absorption peak at 480 nm. The percentage release was calculated based on the calibration curve (Appendix A). The DOX-loading capacity (DLC) and efficiency (DLE) of SeSe_0.01_-PAA-TPE_0.02_ were calculated to be 5.1% and 62.1%, respectively.

Redox-triggered drug release studies in vitro were investigated at pH = 7.4 (PBS, 1 M) and 37 °C by using 0.001 wt.% H_2_O_2_ and 10 mM DTT, respectively. The blank test was to observe the release process without redox stimulation. The release behaviors were monitored through the increase of the UV-vis absorption intensity at 480 nm in solution. As summarized in Figure 4a, the release of DOX was slow and sustained, and only 20% was released from the polymer gels after 40 h due to the blocking effect of the intermolecular hydrogen bond. In the presence of 0.001 wt.% H_2_O_2_ solution, the percentage release increased gradually with time and reached a maximum value (about 60%) within 6 h. A slower release trend could be observed when the polymer gels were immersed in 10 mM DTT and the loaded DOX release reached 25% in 20 h and about 40% in 40 h. This might be attributed to the reduction and oxidation responsive cleavage of diselenide groups oxidized into seleninic acids by H_2_O_2_ and reduced to selenols by DTT [33], which destroyed the cross-linked networks and caused quick release. Meanwhile, the fluorescence emission based on TPE (λ_ex_ = 340 nm) of the DOX-loaded gels in a PB solution was tested during the redox release, as shown in Figure 4b. In the presence of H_2_O_2_ or DTT, the fluorescence intensities in reaction solutions distinctly increased with the loaded DOX release, suggesting a visual drug release process. These results further elucidated the potential feasibility of the polymer gels as a targeted delivery system of hydrophobic drugs.

### 3.4. Cytotoxicity Tests

In the drug delivery field, one of the most important issues that researchers are concerned about is the cytotoxicity of the system. To understand more about our system, the cytotoxicity of SeSe_0.01_-PAA-TPE_0.02_ after the reactions with 0.01 wt.% H_2_O_2_ or 10 mM DTT was evaluated using HeLa cells and the CCK test. It should be noted that 0.5 mg mL^−1^ was the highest concentration used in all the experiments. The results showed the lower cytotoxicity of SeSe_0.01_-PAA-TPE_0.02_ (Figure 5) when compared with other selenide-containing polymers [42,43]. Even if the concentration of degradation products reached 0.8 mg mL^−1^, the cell viability was still around 90%, so the diselenide-linked polymer gels had good biocompatibility.

## 4. Conclusions

In conclusion, we successfully prepared AIEgen-functionalized, diselenide-crosslinked polymer gels, SeSe_y_-PAA-TPE_x_, with both redox responsiveness and AIE effect. The degradation behaviors of the polymer gels and their fluorescence enhancement in reaction solutions under different H_2_O_2_ or DTT concentrations were studied. Furthermore, the polymer gels could be used for controlled drug release. Under the action of external redox agents, the fluorescence intensities in reaction solutions distinctly increased with the loaded DOX release, suggesting a visual drug release process. Therefore, the AIEgen-functionalized, diselenide-crosslinked polymer gels showed great potential for application as biomedical materials for biosensing and controlled drug delivery.

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
