# Peer review of "Novel AIEgen-Functionalized Diselenide-Crosslinked Polymer Gels as Fluorescent Probes and Drug Release Carriers"

_polymers, 2020, doi:10.3390/polym12030551_

Round 1

Reviewer 1 Report

In this manuscript, the authors have studied stimulus-responsive polymer materials, which are oxidation and reduction regulator. Moreover, the polymer material can also perform drug delivery and can control the emission of fluorescence. Overall the study is intriguing to hold the attention of the vast reader. However, the authors have to clarify the following question before accept:

  1. The polymer materials size distribution and stability related study is the absence in this current submission. The authors have also included the PDI data in the manuscript.
  2. The experimental condition needs clarification in Fig 3 (a, b) caption. For example, the excitation wavelength.
  3. Fig 3 (c) caption needs additional explanation to clarify red and black line FTIR peaks. Each FTIR peak has to be assigned to know the change.
  4. In drug release behavior (3.3 section), the authors should add more explanation of how both oxidation and reduction in the same drug carrier could help the release of drug in a designate site of action.
  5. In Fig 3., the authors need to clarify the meaning of the blank sample. In addition to that, the author may also add the release of free Dox as a control group.
  6. The cytotoxic effect may vary cells line to cell line. Different cell lines have a different response in the presence of the same materials. The ref [42] studied the MDAMB cell line that is different from the HeLa cells line. The author should take into consideration this factor.
  7. Furthermore, in ref [42], explore a much lower concentration than the current submitted manuscript (see figure 7 of ref 42, X-axis (μg/mL). So, the authors have to rewrite and re-discussed their finding in Fig. 4

Author Response

Response to Reviewers

Reviewer 1

Comments to the Author

In this manuscript, the authors have studied stimulus-responsive polymer materials, which are oxidation and reduction regulator. Moreover, the polymer material can also perform drug delivery and can control the emission of fluorescence. Overall the study is intriguing to hold the attention of the vast reader. However, the authors have to clarify the following question before accept:

Q1: The polymer materials size distribution and stability related study is the absence in this current submission. The authors have also included the PDI data in the manuscript.

A1: We would like to thank the reviewer for the comments. In this paper, we designed and prepared polymer gel with diselenide cross-linker. The polymer material was a stable three-dimensional network structure formed by chemical crosslinking. It would be swollen and not dissolved in hydrophilic solvents, so it could not be characterized by GPC. In our experiments, the gel samples are particles with a diameter of about 2 mm in dry state.

Q2: The experimental condition needs clarification in Fig 3 (a, b) caption. For example, the excitation wavelength.

A2: Thanks, we have made changes in the original text. The caption of Fig 3 (a, b) has been modified as “Fig. 3 (a, b) Photographic images of SeSe0.03-PAA-TPE0.02 before and after the reactions with 0.01 wt% H2O2 (a) or 10 mM DTT (b) and the additions of Ca2+ under UV lamp (365 nm)”. And test conditions have also been specified for the fluorescence emission spectrum as “λEx= 340 nm, slit width for Ex. & Em.= 5.0, PMT= 400V”.

Q3: Fig 3 (c) caption needs additional explanation to clarify red and black line FTIR peaks. Each FTIR peak has to be assigned to know the change.

A3: Thanks, we have made changes in the original text. The caption of Fig 3c has been modified as “(c) FT-IR spectra of SeSe0.03-PAA-TPE0.02 before (black) and after (red) the recovery”. In the main text corresponding modification has been added as “FT-IR spectra (Fig. 3c) showed that the absorption peak located at 1452 cm-1 appeared belonging to the stretching vibrations of C-Se groups, and the other absorption peaks were also almost consistent before and after the recovery.”

Q4: In drug release behavior (3.3 section), the authors should add more explanation of how both oxidation and reduction in the same drug carrier could help the release of drug in a designate site of action.

A4: Thanks, we have made changes in the original text. “The redox process is constantly and widely present in physiological environments; normal cells keep a lower level of redox agents while redox process in tumor cells would increase significantly [12-14]. So the polymer gels would be used in responsive to redox stimuli in the tumor microenvironment as a targeted delivery system.”

Q5: In Fig 4., the authors need to clarify the meaning of the blank sample. In addition to that, the author may also add the release of free Dox as a control group.

A5: Thanks, we have made changes in the original text. “The blank test was to observe the release process without redox stimulation”. In addition, the release of free DOX is equivalent to a dissolution process, which is very fast and not necessary to prove by experiment.

Q6: The cytotoxic effect may vary cells line to cell line. Different cell lines have a different response in the presence of the same materials. The ref [42] studied the MDAMB cell line that is different from the HeLa cells line. The author should take into consideration this factor.

A6: We fully agree with this comment and added some new references to the article (ref. 42-43).

Q7: Furthermore, in ref [42], explore a much lower concentration than the current submitted manuscript (see figure 7 of ref 42, X-axis (μg/mL). So, the authors have to rewrite and re-discussed their finding in Fig. 4

A7: Thank you for your comment to further improve the manuscript. We have to admit previously inappropriate references, and we have made changes in the original text (ref. 42-43).

Reviewer 2 Report

Zhao et al proposed AIEgen-functionalized Diselenide-Crosslinked Polymer Gels as Fluorescent Probes and Drug Release Carriers. Manuscript is well written and author discussed it results well.  

However, I have few comments.

  1. In IR spectra b) samples II and III has new peak appearing at around 1000 cm-1. Did author characterize those peaks? Moreover, peak at 1728 cm-1 almost disappeared in spectra II and III.
  2. Does treatment with DTT for 24h was not enough to break all Se-Se bonds? because in XPS spectra we still see peak for both Se=O, and Se-Se?
  3. Page 7. Line “The diselenide bond also could be under reduced to selenol reductive conditions” is not clear.
  4. It is not clear and even not much clearly explained why DTT reduced polymer release drug slowly in comparison?
  5. If polymer can be degrading from both approaches the same way, the release of drug should look similar. However, this is not the case. Is it possible that generation of additional hydrogen bonding group in polymer SiOOH, stops quick release of drug?
  6. Fig. 4; Various concentration of what?
  7. If the particles are prepared to release drug in-vivo, how the degradation of polymer can be initiated in-vivo? Please discuss this. In last, how this Se-Se based procedure can be novel if it already studied in past.

Author Response

Response to Reviewers

Reviewer 2

Comments to the Author

Zhao et al proposed AIEgen-functionalized Diselenide-Crosslinked Polymer Gels as Fluorescent Probes and Drug Release Carriers. Manuscript is well written and author discussed it results well. However, I have few comments.

Q1: In IR spectra b) samples II and III has new peak appearing at around 1000 cm-1. Did author characterize those peaks? Moreover, peak at 1728 cm-1 almost disappeared in spectra II and III.

A1: Thanks for your valuable comments. The typical C-Se vibration signal could be observed at 800 cm-1 in the sample I. Such signal disappeared in IR spectra of samples II and III due to the transform of C-Se group. The peak at 1728 cm-1 and 1000 cm-1 also should related with such transformation of C-Se groups.

Q2: Does treatment with DTT for 24h was not enough to break all Se-Se bonds? because in XPS spectra we still see peak for both Se=O, and Se-Se?

A2: Thank you for your comment. In XPS spectra, the peak of Se-Se did not completely disappear after the treatment with H2O2 and DTT. We have also tried to extend the treatment time, but the results were the same. We guess some Se-Se groups were wrapped inside the polymer chains not cleaved. In the previous reports (the original ref. 31 and 34), the results also showed that the peak of Se-Se groups has not completely disappeared.

Q3: Page 7. Line “The diselenide bond also could be under reduced to selenol reductive conditions” is not clear.

A3: Thanks, we have made changes in the original text. “The diselenide bond also could be reduced to selenol under reductive conditions.”

Q4: It is not clear and even not much clearly explained why DTT reduced polymer release drug slowly in comparison?

A4: Thank you for your positive comment. In this report, oxidation and reduction responsive behaviors of the polymer gel under different H2O2 or DTT concentrations were studied, we found that oxidative degradation process was much faster than reductive degradation process.

Q5: If polymer can be degrading from both approaches the same way, the release of drug should look similar. However, this is not the case. Is it possible that generation of additional hydrogen bonding group in polymer SiOOH, stops quick release of drug?

A5: Thank you for your helpful comment. DOX is a hydrophobic drug, loaded into the networks of the polymer gels. The redox responsive cleavage of diselenide groups would destroy the cross-linked networks, and caused DOX release gradually. The release rate is closely related to the cleavage rate of diselenide groups. And degradation products of gels are dissolved in water, SeOOH groups are more likely to form intermolecular hydrogen bonds with H2O (Angew. Chem. internut. Edit, 1969, 8, 499-509).

Q6: Fig. 5; Various concentration of what?

A6: Thank you for your positive comment. In Fig. 5, the concentration is mass concentration of the polymer gels in PB solution before redox degradation.

Q7: If the particles are prepared to release drug in-vivo, how the degradation of polymer can be initiated in-vivo? Please discuss this. In last, how this Se-Se based procedure can be novel if it already studied in past.

A7: Thank you for your positive comment. We know that the redox process is constantly and widely present in physiological environments, normal cells keep a lower level of redox agents while H2O2 and DTT concentrations in tumor cells increased significantly. So the polymer gels would be used in responsive to redox stimuli in the tumor microenvironment.

Diselenide species are of interest as they are active to either oxidation as well as reduction. In 2010, Xu H. et al. reported the first redox-responsive spherical micelles containing diselenide functional group (the original ref. 34). After that, more diselenide-containing drug delivery systems were developed, including micelles, hydrogels, and metal-organic frameworks (MOFs) in response to redox stimuli (the original ref. 30 and Nano Today, 2015, 10 (6), 717−736).

Round 2

Reviewer 1 Report

The editor may consider this manuscript to accept for publication.

Reviewer 2 Report

Authors response to my comments are satisfactory.